# Insights into the Experiences of Treatment for An Eating Disorder in Men: A Qualitative Study of Autobiographies

**DOI:** 10.3390/bs7020038

**Published:** 2017-06-16

**Authors:** Priyanka Thapliyal, Deborah Mitchison, Phillipa Hay

**Affiliations:** 1Translational Health Research Institute, School of Medicine, Western Sydney University, Building 3.G.P9, Locked Bag 1797, Penrith NSW 2751, Australia; p.hay@ westernsydney.edu.au; 2Centre for Emotional Health, Department of Psychology, Macquarie University, Sydney 2109, Australia; deborah.mitchison@mq.edu.au

**Keywords:** eating disorder, men, treatment, experiences

## Abstract

Eating disorders are increasingly recognized as a problem for men but help-seeking is low and little is known about their treatment experiences. This paper sought to determine the treatment experiences of men who have suffered from an eating disorder using autobiographical data. Inclusion criteria were autobiographies of men who had experienced an eating disorder and sought any form of treatment for this, written in the English language, published between 1995 and 2015, and available for purchase in 2016. The search resulted in six books that were thematically analyzed. Analysis of data resulted in two broad themes (1. Positive experiences; 2. Negative experiences) with sub-themes. With regards to the first theme, factors such as concern of staff members, therapist’s expertise (in treating eating disorders in men), and a collaborative treatment approach were considered favorable for treatment. In contrast to the first theme, apathy of staff members, the authors’ own negative preconceptions, treatment providers being perceived as prioritizing financial concerns, perceived as incompetent and judgmental behavior of therapist(s), and time limitations of sessions were considered unfavorable treatment experiences. In this study, the perceived success of treatment depended on therapist’s features and the form of treatment provided. Further research examining these is indicated.

## 1. Introduction

Eating disorders include anorexia nervosa (AN), bulimia nervosa (BN) and binge eating disorder (BED). AN is a condition in which an individual practices extreme food restriction to the point of self-starvation and excessive weight loss. By definition the person is underweight for their age and height. Two subtypes are defined: the restrictive/compulsive exercise and binge-eating/purging types [1]. BN is characterized as a condition in which an individual has episodes of overeating accompanied by a sense of loss of control (binge eating). These are followed by repeated compensatory behaviors that include self-induced vomiting, laxative misuse, excessive exercise, fasting, use of diuretics, and/or other medications to prevent weight gain [1]. BED is characterized as having recurrent and distressing binge eating episodes which may be characterized by eating more rapidly than normal, eating until uncomfortably full, eating large amounts even when not hungry, eating alone because of embarrassment and/or feeling disgusted or guilty after eating [1].

Despite the popular view of these as conditions suffered by women, eating disorders are not rare among men. Raevuori et al. [2] recently conducted a review of eating disorders in men. The paper emphasized the revised Diagnostic and Statistical Manual of Mental Disorders (DSM-5) published in 2013 as an important landmark in recognizing BED alongside AN and BN [1], a disorder almost as prevalent in adult men as in women. Further, the elimination of amenorrhea as a criterion improves conceptual inclusiveness of men with a diagnosis of AN. The overall prevalence of eating disorders in male and female Swedish twins born between 1935 and 1958 was 1.20% in women and 0.29% in men [3] and the lifetime prevalence in Finnish twin men born between 1975 and 1979 was 0.24% [4]. Carlat et al. [5] also conducted a review of prevalence literature and concluded that men account for 10–15% of all bulimic patients, and that 0.2% of all adolescent and young adult men meet strict criteria for BN. Similar prevalence figures have been reported for male patients with AN [6,7]. Hudson et al. [8] conducted a large population-based household survey in the US in 2001–2003 and estimated the lifetime prevalence of DSM-IV AN, BN, and BED as 0.3%, 0.5% and 2.0%, respectively, among men. A similar pattern to that seen in women, with BED prevalence higher than BN which in turn is higher in prevalence than AN, has been reported in men [9]. Despite this, men are excluded from the majority of epidemiological and other research studies, reinforcing the perception of eating disorders as a problem predominantly concerning women, and making it less likely for researchers to design future studies representative of men [10].

There may however be differences between men and women in regards to eating disorder epidemiology. For example, as reported by Mitchison et al. [11] the prevalence of extreme dietary restriction and purging behaviors are increasing at a faster rate in men (2008 vs. 1998 odds ratios = 4.9 and 6.2, respectively) compared to women (odds ratios = 1.7 and 0.9, respectively). While treatment approaches are based on interventions developed for women with eating disorders; there are no male-specific treatment guidelines [12]. Although treatment in predominantly “feminine” environments can be successful for men [13], issues of stigmatization and isolation have also been reported [14,15]. Men may have unique needs, including consideration of masculine identity, gender role conflict and emasculation [16,17]. Further, behavioral manifestations may differ (e.g., more emphasis on muscle-building and exercise in general).

Seeking help for an eating disorder is not common in general, however men with AN and BN have said that their willingness to seek professional help has been further thwarted by societal ideals that include notions of gender-specific illnesses [18]. A quantitative study by Ming et al. [19] evaluated service utilization in men with eating disorders in Singapore. The study reported that a significant proportion of men with an eating disorder had never been hospitalized (62.5%) or enrolled in specialized inpatient/outpatient treatment programs (68.1%). The low rate of treatment-uptake was associated with unwillingness among the men in this study to seek help.

In regard to men who do access treatment, a quantitative study [20] has reported on the experiences of 334 men in a residential center (23% binge/purge type, 24% BN, 23% EDNOS including n = 6 with BED). Overall treatment was reported to be effective. The authors surmised that the men’s experiences of treatment did not appear to differ to that observed among women, but no specific similarities were reported. There were also distinctive aspects of the program perceived as particularly relevant to men. These included addressing excessive exercise, body image concerns with muscularity, sexual orientation, sexual identity, and spirituality. This study also emphasized the importance of “male only” group therapy to facilitate discussion of eating disorder symptoms that have been viewed as “female” problems.

In a recent systematic review [21] of qualitative studies exploring the treatment experiences of men with eating disorders, only four papers were eligible for review. Of the four included studies [22,23,24,25] only two had a primary focus on men’s experiences of treatment [23,24]. Common themes that emerged across the studies were most often related to an understanding of the barriers to help-seeking with less about processes once treatment engagement had occurred. Key themes identified included: (1) delays in seeking treatment; (2) clinical features distinctive to men such as drive for muscularity; (3) feminine and other aspects of treatment services; and (4) a lack of consensus in views about the relevance of gender in treatment. Similarly, in some of the studies the participating men described feeling disappointed with the competency of the help they received and also expressed a wish for their maleness to be recognized when in the treatment setting [26,27].

Similar to studies of females [28,29], two studies [26,27] reported that some males perceived their treatment experience as being unsupportive, with professionals lacking time and experience. Men also reported feeling a loss of power and control when in treatment. The males in this review drew attention to a lack of care providers’ appreciation of male issues. Male participants with AN or BN reported that female-oriented treatments for eating disorders were not entirely appropriate for them.

General stigmatization of eating disorders also affects men’s treatment experiences. Robinson et al. [24] explored the experiences of men using eating disorder services and identified that the biggest challenge they had faced was admitting to themselves and others that they had an eating disorder and that it was a problem. This may be because of the stereotype that eating disorders only affects women. Similarly, a study by Griffiths et al. [30] supported this premise, as males with eating disorders reported that they were being more frequently stigmatized as “less of a man” than women experienced being stigmatized as “less of a woman”.

## 2. Deficits in Previous Research

Whereas quantitative research aims to assess causal relationships between variables, qualitative research understands truth to be evolving, culturally constructed and derived from an interaction between experience and the human mind [31]. Qualitative methods are particularly suited to understanding a lived experience, as they target underlying processes and meaning that cannot be adequately conveyed in surveys or represented by numbers.

Qualitative literature pertaining to men’s experiences of eating disorder treatment is scarce. Historically few men have been included as participants in such research designs. In a systematic review and meta-synthesis of qualitative studies [32] on patients’ understanding of eating disorder treatment and recovery, the participants were almost exclusively women. Likewise, Bezance and Holliday [33] identified 11 qualitative studies of adolescents’ experiences that included only one paper with a single male participant. In the eating disorder field men are either not being represented at all, or at best, studies have included only very small numbers [34,35,36,37]. Other limitations of previous qualitative studies include a narrow focus on a specific eating disorders or subset of eating disorders, such as AN [38] or AN and BN only [39]. This is problematic as we know that men are more likely to experience disorders such as BED, and are over-represented in the residual diagnostic category, Other/Unspecified Feeding or ED. To our knowledge there is no qualitative study to date that has focused primarily on treatment experiences of men with an eating disorder. 

## 3. Aims

Eating disorders in men is an under-researched area, particularly with regards to the exploration of male experiences in treatment. The aim of this present study was to explore the experiences of men who ever had any form of treatment for an eating disorder. Given the relatively unexplored nature of this topic, a qualitative approach was deemed most suitable. 

## 4. Methods

### 4.1. Study Sample: Men’s Eating Disorder Autobiographies

The sample consisted of autobiographical accounts written by men who had experienced an eating disorder. Inclusion criteria were that the book should be an autobiography of a man, who had or has had a treatment experience with an eating disorder. Further, the books were to be written in the English language and published between the years1995 and 2015.

### 4.2. Autobiographical Literature Search

Autobiography is referred to as a form of “personal document”. At the core of personal document research is the life story—an account of one person’s life in his or her own words. For example, oral interviews, testaments, literary biographies, and psychological case studies bring life stories into being that would not otherwise have happened in everyday life [40]. On the other hand, personal documents include “any first-person narrative that describes an individual’s actions, experiences and beliefs” [41]. Autobiographies provide a complementary and rich source of qualitative data regarding authors’ attitudes, beliefs and views of the world. It reflects the author’s perspective, which is what most qualitative research is seeking, and it also provides a subjective account of the situation it records. In addition, autobiography as a source of unsolicited data in our view may reduce one element of investigator bias when compared to other commonly used forms of qualitative data, such as participant interviews, in which the investigator chooses the topic and questions to be posed. Aronson [42] analyzed around 270 book length stories, most written in the past 20 years, and concluded that more men than women write about their illness. He concluded that men write out of a desire to help other patients, and to come to terms with their own illnesses, and also for doctors to have a better understanding of their experiences. Thus men may primarily write to express repressed emotions and create awareness about male mental illness, and less often are driven to write their autobiography out of mercenary motivations.

The search for published autobiographies was conducted using Amazon.com and the keywords were [“biography” AND “man” AND “eating disorders”]. The search resulted in twelve books. Further scrutiny of the texts led to the exclusion of six books: one was written by a mother, two were concerned with recovering from obesity rather than an eating disorder, one was about the presentation of illness narratives, one did not mention treatment experience, and one was a collection of motivational stories on staying young. Additional searches of Booktopia and The Book Depository produced 4 and 31 books respectively, of which none were autobiographies of men with an eating disorder.

## 5. Procedure

The qualitative method employed was thematic analysis, according to the five phases in the Framework method outlined by Ritchie and Spencer [43] and Pope, Zeibland and Mays [44]. This comprised the following stages:

(1) *Familiarization* (reviewing a subsample of the raw data in detail). This involved the first (PT) and second (DM) authors reading the books two times completely to familiarize themselves with the content and to empathize with the authors.

(2) *Identifying a Thematic Framework*. This involved ascertaining all key issues and themes in the books by drawing both from the research aims and from the data itself. PT and DM undertook this separately and then discussed the themes identified until a consensus was reached on an overarching framework.

(3) *Indexing*. This phase was undertaken by PT and involved systematically applying the agreed upon thematic framework to the data by coding sections of text. Any newly emerging themes were identified and discussed at regular meetings with the other authors, and if agreed, added to the framework.

(4) *Charting*. This involved rearrangement of distilled summaries of the data according to the thematic framework. A thematic tree was formed to highlight the main themes that were identified by the first author (PT).

(5) *Mapping and Interpretation*. This involved finding associations between themes and mapping the range and nature of phenomena to find explanations for the findings. This was carried out by PT and DM with regular review by PH.

## 6. Results

### 6.1. Characteristics of Sample

The autobiographies of six men that suffered eating disorders formed the data sample. The age of authors ranged from 25 to 50 years. A brief description of each book follows and is also presented in Table 1.
(1)“Born Round” [45] presents a sufferer’s account of binge eating and bulimia nervosa. In this book Frank Bruni describes his helplessness as a man with an eating disorder. Although he realized that he had a problem, it took him years to get help, due to self-stigma and shame. Frank eventually sought help from a therapist, the process of which he describes, and moves forward toward living a life free of binge eating.(2)“Hiding Under the Table” [46] is authored by Dennis Henning who suffered from BED, BN and AN. This book is written from a recovered perspective and focuses on the barriers experienced in trying to access treatment. Dennis provides an account of his misery as a man who had eating disorder. He desperately tried seeking treatment but often found his gender denied him access. After recovering from his eating issues Dennis pursued his goal to create public awareness about eating disorders. He is also co-founder for the Lifestyle Institute for Eating Disorders.(3)“Weightless” [47] is an autobiography written by Gary A. Grahl that presents a sufferer’s perspective of BED. In this account Greg moves through the process of illness and recovery. Presently, he is involved in creating awareness through his blog JustStopEatingSoMuch.com, which focuses on weight loss and food addiction.(4)Michael Krasnow authored the autobiography “My Life as a Male Anorexic” [48] from a suffering perspective. He suffered from severe AN and wrote with the intention to help other young people, as well as to assist health professionals to better understand and provide help to sufferers. His account is almost exclusively focused on his experiences in treatment. Michael died three days after writing the final epilogue of this book.(5)“My Thinning Years” [49] was written by Jon Derek Croteau. In this book he presents an account of how his denial of his own homosexuality led to the development of AN, BN, and other obsessional behavior.(6)Gary Grahl wrote “The Skinny Boy” [50] and now is a professional counselor in Wisconsin and also a resource person for ANAD (National Association of Anorexia Nervosa and Associated Disorders) in Illinois. The narrative of Skinny Boy is about a young man suffering from AN. In his book Gary described in detail the daily struggles of living with AN and experiences of treatment that he had in the inpatient psychiatric unit of a hospital.

Three of the autobiographies were about men with a history of AN, of which two (The Skinny Boy & My Life as a Male Anorexic) provided details of experiences within eating disorder specific inpatient treatment settings. The other autobiography (My Thinning Years) discussed experiences of outpatient Cognitive Behavioral Therapy. BED was reported in two of the books, where one author discussed outpatient therapy sessions (Weightless) and the other (The Good Eater) did not describe the type of treatment experienced, excepting that it included self-help strategies such as “changing mindset” and exercise. Only one (Hiding Under the Table) author identified himself as suffering from BN and he describes his experiences seeking and receiving treatment from an inpatient rehabilitation center for eating disorders. All of the autobiographies described eating problems that first emerged during childhood. Two of the six authors identified themselves as homosexual. All autobiographies included rich detail about the authors’ experiences of suffering from an eating disorder, and all but one included details about the authors’ journey to recovery from their eating disorders. Treatment experiences were described in detail in 4 books, where text about treatment appeared on at least 15% of pages. This included two books (The Skinny Boy and My Life as a Male Anorexic), which were almost entirely about the treatment experienced by the authors. The remaining two books focused more about the reasons and situations that led to the eating disorder and discussion of treatment appeared on around 5% of all pages.

The analysis of data resulted in two broad themes. (1) Negative experiences of treatment and (2) Positive experiences of treatment, under which emerged several sub-themes.

### 6.2. Negative Experiences of Treatment

The authors provided many reasons for a negative experience. These included the behavior of the treating staff, negative thoughts and preconceptions of the men themselves, the time constraint of treatment sessions and also a lack of expertise of the health professionals.

*(1)* *Unhelpful Interactions with Health Professionals*

*Judgmental.* Firstly, the perceptions of authors about health professionals being invasive emerged as an important issue, though it was not the case every time. The authors identified that it was quite challenging for them to visit a health professional due to what was perceived as a stigmatized and judgmental stance against the patient. For example, one author mentioned that whenever he visited the doctor he felt as if he was being judged, which heightened his self-consciousness about his weight.
“I had finally gotten around to selecting and paying a visit to a doctor. I’d neglected it before because I always avoided doctors, whose poking and prodding and above all weighing of me amounted to a judgment I didn’t want rendered. Doctors made you stand naked or half naked in front of them.”[45] (p. 203)

In the quote above, the author refers to feeling “naked” in front of the doctor, which may be interpreted both literally as well as metaphorically as being exposed. The “prodding” and “poking” referred to also suggests that rather than feeling accepted and understood by the health professional during such exposure, the author felt threatened and critically examined. The attitude of a therapist as being judgmental about the men’s past also emerged as a common issue. As described in the account below, the author felt that his own healthy intentions toward therapy and recovery were being thwarted by the judgmental nature of the therapist:
“I entered an eating disorder clinic. The therapist told me I had to really work on things and share in groups and to take my medications. I felt nervous but open. I rebelled against some groups and a yoga class, so the doctor came to visit me one day. He was very judgmental about my past and what I had done to survive.” [46] (p. 65)

*Unfriendly Attitude of the Therapist.* A few authors described an “unfriendly” attitude of therapists toward them, which was thought to sometimes emanate from an antipathy against males. This type of attitude in a clinical setting plays a particularly important role because the treatment provided, as well as the patient’s response to treatment, are influenced by the preconceptions of the therapist, as well as those of the patient.
“There are people and therapists who want to help others, but who cannot even help themselves. There was this therapist who had issues with men that are not resolved and she became a therapist to help women who suffered like her. It is sad because they still harbor anger toward men and while treating men they are not able to really separate their anger.” [46] (p. 83)

Another example of unfriendly and stern behavior is exemplified in this quote by another author, below. This author felt that his attempts to build rapport with his therapist were unsuccessful, which negatively affected the therapeutic relationship.
“I also didn’t like that he never laughed at my jokes and that he always had a stoic expression on his face whenever I explained some of the circumstances of my upbringing. Where was this guy’s compassion? Didn’t he knows I was a victim and unable to help myself? That none of this was my fault?” [47] (p. 163)

*(2)* *Thoughts and Behaviors Impeding Treatment*

Apart from the attitude and behavior of health professionals, the mindset of the person in treatment is very crucial for a positive outcome. The analysis of data found that sometimes it was the preformed notions of patients that negatively affected treatment outcomes, such that they were slow to engage in therapy. 

*Low expectations of therapists.* Authors sometimes described unhelpful negative beliefs about therapy and therapists that influenced their attempts to seek and engage in treatment. The following excerpt from an autobiography indicates the self-assertive thinking of an author along with denigration of the health professionals.
“I went into rehab knowing—and I mean knowing—that I knew it all and that the professionals there did not know jack shit.” [46] (p. 177)

Similarly, an unsparing disclosure by one of the authors demonstrated the extent of the negative perceptions of treatment providers that may occur. The author in the following excerpt claims that psychiatric help is of no use.
“Psychiatrists and psychiatric hospitals are pointless, if not detrimental.” [48] (p. 52)

Such beliefs are likely formed on the basis of prior experience, however clearly place individuals in a position “against” rather than “in alliance” with the treatment team even before treatment commences.

*High expectations of therapists.* In contrast to the low expectations of the competencies of health providers, sometimes it was the patients’ overly high expectations of health professionals, and the health professionals’ subsequent inability to fulfill such expectations, that led to negative treatment experiences. An author, who was severely ill with AN in particular used the term “miracle workers” for therapists/doctors and reported that while initially he thought highly of doctors, over time, as he did not recover, they were viewed less favorably.
“To us doctors were miracle workers. We had the incorrect belief that when you go to a doctor with your problem, he can solve it with a snap of his fingers.” [48] (p. 11)

This alternating idealizing and demeaning of health professionals may be symptomatic of an unhealthy attachment or interpersonal style, which serves to create strain and distance in the therapeutic relationship, and thus hinder recovery.

*Self-sabotaging.* The authors of the autobiographies wrote that they felt that they were being judged within treatment centers, and were not well understood by others, including staff and other patients. This often resulted in unhelpful or self-sabotaging behavior in an attempt to have their need for acceptance met. In one instance an author recalled how he intentionally lied to a therapist in order to gain acceptance from him:
“Jeff asked me to tell the truth during our first meeting and I thought that even though I was in a rehab and they were there to help me, he would think that I was a freak, a loser, unworthy of love, a pathetic excuse for a human being. I thought I had to lie to him to be accepted, so I lied.” [46] (p. 167)

*(3)* *Therapist Limitations*

In continuation to the first theme of author’s perception about the treatment providers’ behavior, their knowledge regarding eating disorders and their expertise in the field as perceived by the men in this study emerged as a significant issue. All the authors unanimously acknowledged the expertise of therapists as an important factor in determining their treatment outcomes.

*Lack of expertise in treating eating disorders.* Perceived incompetency of therapists emerged as an important and critical issue in the study. After a person becomes sure that he is suffering from a health problem, the primary concern becomes the need for it to be resolved. In the case of an eating disorder this is not always simple, as exemplified in the following quote in which an author reported the inadequacy and unavailability of accurate treatment in men:
“Some places tell me that they have a program for men, but the truth is they have no idea how to work with men, nor do they understand the differences between men and women who had eating disorders.” [46] (p. 80)

The same author further added that the claims made by treatment centers that they were capable of treating eating disorders in men were false and there is a notable scarcity of centers that know how to treat a man with eating disorder.
“Until this stay I had seen eight different professionals who tried to help me and at the same time only two knew how to work with someone who suffered from an eating disorder.” [46] (p. 67)

Contradictory to his expectations that treatment providers would understand eating disorder issues, one author reported feeling misunderstood by the staff. Perceptions that eating disorders occur in women only may explain such disregard of the male author’s symptoms.
“The doctor and nurses made me feel as if I was crazy. None of them asked why I’d dropped so much weight.” [49] (p. 172)

The experience of rehabilitation for men varied. One of the authors described his rehabilitation stay as a “real eye-opener” because of the conduct displayed by the staff members.
“The one thing that was really an eye opener was to me in my last stay in rehab is that the people treating me were no healthier than I was.” [46] (p. 67)

*Ignorance of eating disorders in men.* Furthermore, another author declared the therapist as ignorant and unaware of the occurrence of eating disorders in men. Interestingly, this author, who was severely underweight at the time, discussed how even though he was displaying excessive distress about fatness, the health professionals did not assess him for an eating disorder. To this author, such an experience spoke to the dismissal of the fact that eating disorders exist in men:
“I was freaking out about how much fat was in the intravenous fluids they were giving me, but ironically, it never occurred to anyone to ask whether I had any kind of eating issues. It probably never occurred to them that a male could be anorexic. Even though I told the doctor I was running twelve miles or more a day on a bowl of cereal or a bagel and a handful of fat burners, he never asked about an eating disorder.” [49] (p. 172)

*Money-mindedness.* Half of the men mentioned the money-mindedness of therapists. They quoted many instances where they felt their individual needs were neglected because the therapist prioritized monetary gains. This finding was similar, irrespective of the place of treatment (e.g., residential care, inpatient setting, or counseling session). The following excerpt clearly stated the frustration and misery of an author in this regard:
“He was getting paid around sixty dollars per hour and we were staring at each other. That’s all. No talking. I frequently told my parents that I wanted to stop seeing Dr P. He did not help and was a complete waste of time. I could be wrong but I believe he was more interested in money than anything else (either that or he was totally incompetent).” [48] (p. 24)

*Ineffective Communication.* Staff members have a critical role in the treatment setting of a clinic or hospital. According to the majority of authors the staff members were expected to be more authoritative, as opposed to authoritarian, in order to bring a subtle and long-lasting change in their behavior. For example in the following quote the author wished to have a tangible plan for recovery.
“I needed something more concrete, something that was going to hold me accountable and responsible for my behaviors and actions.” [46] (p. 170)

*(4)* *Feeling Ostracized as a Man*

*By staff members.* The role of staff members in the treatment setting was duly acknowledged. Being a man with an eating disorder is itself very challenging. A few men in treatment reported feeling unwanted in the treatment setting by the staff members due to their male gender.
“Some of the staff had a problem with a male in the group and they did not have a problem letting me know.” [46] (p. 67)

*By other patients.* Some men in treatment described feeling misunderstood and targeted by the prejudiced behavior of female patients towards men. For example, one author stated that women in the group could not accommodate his presence and wanted him to leave because they had some personal issues with men in their lives:
“The meetings I first went to were all women and young girls - maybe a man once in a while. In these meetings, I was looked at as an outsider. In one OA (Overeaters Anonymous) meeting, a woman went as far as to ask me to leave because they had issues with men in their lives and a man in group was hard for them to deal with.” [46] (p. 170)

*(5)* *Treatment Side Effects*

Two out of six authors wrote that antidepressants and electroconvulsive therapy were deleterious to their health. In particular, authors discussed unwanted side effects and/or ineffectiveness of medication:
“Dr C prescribed antidepressant medication. Some were stopped because of side effects, such as drowsiness, light-headedness, fidgetiness (the most bothersome - like the jitters), and uncontrollable shaking. Others were stopped because they had no effect one way or the other.” [48] (p. 11)
“Prozac hadn’t given me the speedy buzz it gave some people. It had given me the opposite: a gauzy lethargy.” [45] (p. 200)

One author who described his treatment with electroconvulsive therapy, reported concern regarding memory loss that occurred following therapy:
“One of the possible side effects of ECT is memory loss and this was the most prevalent result of my treatment.” [48] (p. 11)

### 6.3. Positive Experiences of Treatment

The autobiographies focused very seldom on positive, as opposed to negative, treatment experiences. However in contrast to negative experiences, which were at times a result of the author’s own thoughts and behaviors, it is interesting to note that the factors that were considered as positive in treatment were usually external, such as the behavior of staff members in hospital, therapists, or other patients in the treatment center.

*(1)* *Therapist*

*Non-judgmental empathic behavior.* Three of the six authors described non-judgmental behavior of therapists as an important positive experience in treatment. For example, one of the men felt encouraged to continue his therapy sessions because of the uncritical attitude of the therapist. In fact, he admitted that he actually liked being in her company.
“She was the first therapist who took the time to help me and not judge me. I liked seeing her and listening to her.” [46] (p. 92)

When the authors of the autobiographies perceived that the health professional was being empathic, this was important to them in regarding the treatment as positive. This is opposed to the negative treatment experience themes discussed above where the therapist was perceived as being judgmental or unfriendly. Thus, an understanding nature, non-judgmental behavior, and a caring and gentle attitude were therapist qualities that were discussed by authors as contributing to the success of their treatment. The following quote reflects on how a therapist who had a gentle nature was appreciated.
“He had a careful, gentle way of helping me, as if he were carefully removing a bandage that had covered a massive wound for years. He didn’t tear it off; he pulled back the adhesive slowly but steadily.” [49] (pp. 228–229)

*Expertise.* The therapist had a very important role in defining the treatment outcome. Trust in the treatment aligned with the perceived expertise of the health professional, as exemplified in the following quote:
“Once I started to work with the idea that the therapists knew more than I did at the same time, and I began to trust their input in my life, I began to trust myself and how I really felt.” [46] (p. 86)

Interestingly, this theme appears to be in contrast to the negative treatment experience theme where authors described how their preconceptions of the treatment providers’ incompetence at times sabotaged their ability to engage in therapy. Other authors acknowledged the expertise of therapists in helping them to address their symptoms.
“I learned from Jeff R. and Jeff S. how to look at situations and handle them from a positive and realistic point of view, rather than turning to food, sex or the other dysfunctional behaviors I turned to as treatment for my emotional pain.” [46] (p. 167)

*Effective communication style.* The adage that “one rule applies to all” defies for eating disordered patients, as is evident from the varied views presented by the authors regarding the style of communication they preferred from their treatment providers. Whereas some authors described a desire for a “tough” approach from their therapist, others clearly desired a more gentle approach. This demonstrates the need for therapists to assess the communication and relationship needs of their clients in informing their own interaction style. 

Under the “non-judgmental” theme above, an example is provided of an author who appreciated a gentle and slow approach from his therapist: “*carefully removing a bandage*”. On the other hand, for several of the authors, an authoritative nature of the doctor was considered as a welcoming effort to men in treatment. Perhaps this approach instilled confidence in the men, or gave them the permission to relinquish control over their eating to an authoritative figure. The metaphor of the bandage is used again here.
“I have to admit that it was some of the best (and toughest) therapy I’d ever received. I’m not knocking whatever kind of therapy may work for someone else. But for me, this quick, tough love, rip-off-the-bandage-fast approach was really powerful.” [47] (p. 168)
“Dr X was making a major difference in my life. After he had earned my trust, he challenged me and never let me off easy. I liked that he pushed me.” [49] (p. 263)

*(2)* *Effect of Others in Hospital*

*Staff members.* The role of staff members as being friendly and considerate had an important influence in forming the treatment experience as positive. One of the men in this study stated that being able to accept help from staff proved beneficial to him.
“I have learned to let people into my life and trust them to help me, rather than to try to do everything alone and fail.” [46] (p. 188)

*Other patients.* In addition to this, the supportive nature of group members emerged as a positive aspect of treatment.
“We listen. We share pain and heartache. We ask questions. We empathize without feeling sorry for each other. We find the support to heal and gather strength to go on.” [50] (p. 95)

*(3)* *Treatment forms*

There was lack of consensus in men with regards to the most effective form of treatment provided for an eating disorder. 

*Group therapy vs. individual therapy.* In a hospital, the usual form of treatment includes medicine, counseling sessions, group therapy and behavioral therapy. In most of the instances being a part of group therapy was considered as an effective measure by the authors while in treatment. This could be attributed to the fact that talking about their eating disorder openly in group with other members with similar issues provided a sense of normalization and empathy. The following extract clearly outlines the perceived benefits of group therapy:
“Sitting there in groups and slowly discussing my issues with a mind open for feedback made it easier to confront my fears, shame, guilt, anger, low, low, low self-esteem, self-hatred, and lack of self-respect. I no longer believed I had to hide and numb my pain.” [46] (p. 175)

In contrast to this, other authors reported that they felt more supported in individual therapy. For example, in the following excerpt the author clearly stated that he felt safe and secure in therapy sessions where he does not have to acknowledge the presence of others as in group therapy:
“I preferred the quiet safety and the anonymity of the one-on-one therapy.” [49] (p. 213)

This clearly indicates that understanding the individual needs of men is central to tailoring an effective treatment regime.

*Cognitive Behavioral Therapy.* The effectiveness of treatment provided was frequently the primary concern of these men when discussing treatment experiences. In the majority of cases, cognitive behavioral therapy was followed and this approach was perceived as helpful.
“After months of gaining my trust and respect through listening to my story, and understanding my past to help me move forward, she introduced me to cognitive behavioral therapy. This approach and other tools she gave me were meant to help me notice the behaviors I was engaging in and stop them.” [49] (p. 214)

*(4)* *Hospital Environment*

The general atmosphere of the hospital environment was also mentioned as important. Although at times isolating, as reflected by one of the men, this characteristic of hospitalization was appreciated. This may in part be due to a desire to maintain control of the eating disorder, which may be threatened by the interference of others, as expressed here:
“This hospital thing will fit my lifestyle quite nicely since I’ve recently become allergic to people. Social interaction makes me break out in a terrible rash called guilt; I simply can’t seem to please enough people, no matter how hard I try.” [50] (p. 9)

Maintaining a restrictive-type eating disorder requires significant self-control and determination, which is taxing for an individual. Quite a few men mentioned that being hospitalized provided temporary relief from this self-imposed control, which could be relinquished to staff.
“My feelings about the hospital were similar to those I had when I was taken out of the school. Deep down I think that I wanted to be hospitalized. It was a relief. Again it was a control issue.” [48] (p. 14)

## 7. Discussion

The present study examined the treatment experiences of men who had or have had an eating disorder and engaged in any form of treatment. The analysis found that the factors that most commonly contributed to a negative experience were judgmental behavior of professionals and their perceived lack of expertise. The findings regarding the judgmental behavior of health professionals are commensurate with those reported in previous studies by women [28,29]. Additional themes also emerged however, such as the ulterior motives of therapists, like money-mindedness. Though financial cost of treatment had been discussed earlier [51,52,53,54], the putative “mercenary” attitudes of therapists has not been cited previously. Unexpectedly, a few of the men reported an antipathy of female practitioners towards them in treatment. This biased treatment was made sense of by the authors who assumed that some female practitioners had unresolved conflicts with men in their own lives and projected these towards the males they encountered in their treatment centers. All the negative experiences encountered by males thwarted their further engagement in therapy, as has been reported by women in other studies [55]. In contrast, an empathizing nature, non-judgmental behavior, good understanding and expertise with eating disorders in men, and a trustworthy relationship with the health professional formulated a positive experience of treatment. This finding is in consensus with studies in women [28]. 

Interestingly, the study found that sometimes men’s own thoughts and behaviors impacted on treatment experiences. For example, having very high expectations of the therapist as a “miracle worker” led to disappointment when treatment was not progressing. Also, one author reported intentionally misleading therapists at times, to the detriment of their own outcomes. This self-sabotaging behavior may have occurred due to a fear of criticism, or a fear of losing the perceived positive aspects of an eating disorder, indicative of a lack of readiness to change [55]. Also, it was interesting that sometimes different approaches were preferred, for example some preferred a female therapist while others had no issues with the gender of the therapist. This highlights the importance of individualized or person-centered care [24] and increased availability of male providers. 

In treatment, the role of group members and staff was recognized as important by the men in the present study and consistent with men reporting feelings of isolation in prior studies [14]. The study identified that men in the present study felt misunderstood and unwanted by staff members [26,27]. Some authors endorsed the desire for healthcare staff to be authoritative (expressing confidence in their expertise) but to also provide an equal distribution of power and to be empathic and friendly. Feminization of the disorder in society by referring it as women’s disease or a “gay problem” [18] and also the stigma attached to men with eating disorders discouraged the male authors in this study to seek help [24,30]. Also, the sexuality of the authors was not a major theme as only 2 of 6 was identified as homosexual. 

The study supports the limited literature that treatment approaches should consider men also and not only women [12]. However, it should be acknowledged that there are efforts being made to improve the situation and to provide educational material (e.g., videos) to clinicians. Examples include ways to ask questions to men about their eating patterns and exercise as described by Hildebrandt and Craigen [56].

In this study, there was a gender disparity in the number of treatment centers available to the men and there appeared to be a deficit in male-only treatment facilities. The men in treatment had varied opinions about the form of therapy. However, the majority of them promoted group therapy because of the supportive environment experienced within the group, as stated by Weltzin et al., [20]. In contrast to group therapy, there were few that reported positive experiences within individual therapy. Regarding the environment of the hospital, there was no consensus, as some pointed to the “stillness“ of hospital as scary, whereas others preferred the isolation of the unit as a way to escape societal pressures. For these latter authors, the ability to relinquish control to hospital staff members in regards to diet and exercise schedules was experienced as a temporary relief from the pressures of the eating disorder.

## 8. Strengths and Limitations

Strengths of the study include autobiographical accounts of the men with eating disorders about their treatment experiences and broad inclusion criteria of all forms of eating disorders. Further strengths include that a systematic search was employed to identify autobiographies; two investigators read the autobiographies and coded themes, increasing the validity of the emergent themes; and the coding of themes by the authors continued until saturation was achieved. However, this study also has some limitations including: the very small sample size that consisted of a highly specific group of men who (1) decided to write an autobiography about their experiences and (2) found a publisher for it; selection of autobiographies to one language (English); and defined dates (years 1995–2015). It is possible that relevant autobiographies in other languages and/or from different (non-Western) cultures, or other time periods were missed. In addition the study did not include men younger than 18 years of age, so it was not possible to discuss how this age group reflect on their treatment experiences. On the other hand, all authors included in this study wrote about their childhood and adolescent experiences of eating disorder symptoms. In addition, while two authors who identified as homosexual were included in this study, it is possible that the treatment experiences of men with varied sexual identities differ and it would be relevant to include more men with homosexual or transgender identities in future studies. Autobiographies are also limited in that there are unknown motivations for writing (including income from royalties) although research [42] suggests several altruistic motivations for men who write about an eating disorder such as AN. Thus, it would be also important to follow this research with larger samples and with face-to-face interviews that would support the validity of these preliminary findings. Finally, while all authors wrote about treatment experience in the US, people’s experiences may differ in other countries where health care services are government-funded, and where doctors are salaried (e.g., there may be less perceived “money mindedness” of the doctors). 

## 9. Conclusions

This is the first study that used autobiographical accounts of men with eating disorders to understand their treatment experiences. The study identified that the behaviors and attitudes of both treatment providers and male patients may influence the outcome of treatment defining it as a positive or negative experience. Furthermore, the study provides a platform for designing further research and consideration of treatment strategies with a primary focus on attending to patients’ perspectives. In particular, the findings highlight the importance of specialized expertise and a non-judgmental understanding in healthcare providers caring for men with eating disorders. Pre-treatment strategies aimed at educating patients as to expectations of treatment, and practitioner training interventions may be helpful to address barriers to positive treatment experiences.

## Figures and Tables

**Table 1 behavsci-07-00038-t001:** A brief summary of each of the autobiographies analysed in the study.

Book	Author	Country (Year)	ED Diagnosis	Self-Perceived Status at Writing	Sexual Orientation	Occupation	Age of ED Onset; Duration
Born Round [45]	Frank Bruni	US (2010)	Binge Eating, Bulimia Nervosa	Recovered	Homosexual	Restaurant critic	5 years; 30 years
Hiding Under the Table [46]	Dennis Henning	US (2004)	Binge Eating, Bulimia Nervosa, Anorexia Nervosa	Recovered	Heterosexual	Salesperson	9 years; 30 years
Weightless [47]	Gregg McBride	US (2014)	Binge Eating Disorder	Recovered	Heterosexual	Writer & Producer^†^	8 years; 23 years
My Life as a Male Anorexic [48]	Michael Krasnow	US (1996)	Anorexia Nervosa	Not Recovered (Later deceased)	Heterosexual	Library staff	15 years; 13 years
My Thinning Years [49]	Jon Derek Croteau	US (2014)	Anorexia Nervosa, Bulimia Nervosa	Recovered	Homosexual	Consultant, Writer & Speaker ^†^	14 years; 18 years
Skinny Boy [50]	Gary A Grahl	US (2007)	Anorexia Nervosa	Recovered	Heterosexual	Professional ED Counsellor ^†^	15 years; 5 years

NB: ED = eating disorder. ^†^ Occupation following eating disorder recovery.

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
