# Peer review of "Insights into the Experiences of Treatment for An Eating Disorder in Men: A Qualitative Study of Autobiographies"

_behavsci, 2017, doi:10.3390/bs7020038_

Round 1

Reviewer 1 Report

The manuscript ‘Insights into the experiences of treatment for an eating disorder in men: A Qualitative study of autobiographies’ explores 6 autobiographies of men who have had experience as eating disorder patients. The authors sort to ascertain information about the treatment experiences of men with eating disorders. Using thematic analysis the authors attempted to describe the positive and negative experiences of these men. The study is novel in its method and contributes some insight to an area where there is little area on men’s experiences.

Strengths:

1. This study uses an interesting approach to explore the treatment experiences of men with eating disorders.

2. The introduction shows there is a gap in the literature and makes the case that research is needed.

3. With more information on your methods and thick descriptions in your results this paper could provide some useful insights from which to generate another qualitative study into patients currently engaged/disengaged with treatment.

Weaknesses:

If word limits are an issue you could look at making your point about under representation of males in the ED literature more succinctly. At present this point is made repetitively in the introduction.

Research methods:

The research methods have been included after the conclusion. This is unhelpful. Suggest you move to follow the introduction and before results so one can determine how the authors come to their data in the first instance and can make a critical judgement about the results with this context.

It is not clear why autobiographies were chosen as the primary data source

Rationale should be included as to why you only used Amazon as a source of books

Results:

The study has not capitalised on the novel method of analysing autobiographies and has tended to treat the data like one would an interview transcript. This undermines the study as there is little context about the books to allow the reader to be certain that treatment experiences were the focus of the books or were they a side issue that were discussed briefly. The results doesn’t give the reader confidence in the depth of the analysis.

As presented the results are superficial and lack thick description. Thick description is needed given the small sample size. [For example Evidence for theme ‘unfriendly’ is not well substantiated.]

One or two sentences under each theme/category is inadequate to communicate experience well to the reader.

Discussion:

I would suggest you reflect on what the key findings are and discuss only this. Rather than mention every point in your discussion. Avoid introducing new information about the books/your findings/thoughts in the discussion/conclusion.

Limitations:

Limitations downplay significant issue of small sample and also of issues relating to how people write for publication (as opposed to a diary that is not sold for profit) and the impact of this on the data quality.

Specific comments

Spacing needs fixing throughout i.e. sometimes there is no space between sentences or between words.

Line 44: Replace ‘is’ with ‘are’

Line 48: word missing

Line 95-97: Orphan sentence. Join to previous paragraph

Line 170 – what is a ‘representative case’? How do you know it is representative?

Line 179 – you say ‘analysis of themes’. Do you mean analysis of data?

Line 181: “I went into rehab knowing and I mean knowing that I knew it all and that the professionals there did not know jack shit.” (Hiding Under the Table Pg. 177) – you have this quote as self-sabotaging thoughts and behaviours. To me, this is not a very good illustrative quote for this. You have not clearly elicited how this quote reflects the theme you have connected it with.

High expectations – the quote illustrates the patients’ high expectations OF the therapist but your discussion under this heading relates to professionals having high expectations of their patients.

Line 218: who’s expected belief?

Line 227: Is the term ‘participant’ correct in the context of this study? – It seems an odd use of the word given that you bought their book and analysed this. They didn’t participate in your study per se.

Line 285 – you seem to state that there wasn’t much positive to say about treatment yet you have this as a major theme heading. These arbitrary descriptors ‘positive/negative’ seem superficial and reflect categories rather than rich themes.

Line 322 – interesting that this person liked the rip off the bandage fast approach whereas the other quote suggested pulling back the bandage slowly was needed. So different approaches were liked.

Line 357 – ‘definitely the primary concern’ – how could you be certain if you were unable to ask them? Perhaps a little more caution is needed throughout your results to reflect the lack of certainty you can have about the meaning that the person is relaying in their book

Line 382 – It is impossible to know whether they are talking about eating disorder recovery as the insight you have given about the context of each book is non-existent. I can’t see how you can make the leap between them mentioning aspects that were positive or negative about their experiences with treatment in the context of an eating disorder and state that these relate to affecting recovery positively or negatively.

Line 385 – You state – ‘Also the study reported that the perceptions of adult male patients greatly influenced treatment experiences’ I don’t think this statement is well supported by your results

Line 389 – 3 of 6 mentioned money. Given your very small sample using language like ‘repeatedly emerged as an issue’ seems overstated.

Line 411 – Not a major finding to discuss.

Line 423-427 – that people wrote out of a desire to help others etc was not mentioned prior to this point here in the discussion. This seems like an assumption that you have made without sharing with the reader how you have come to this conclusion.

Line 488 and 491 – incorrect tense in this sentence

Author Response

See file attached

Author Response

See file attached

Round 2

Reviewer 1 Report

The manuscript now offers a lot more context to the reader and depth of analysis which has improved the paper significantly.

Major comment:

Much of the discussion reads largely as a summary of the results. For example the section on the type of therapy (group v individual) just states the result. There is a need to go the next step to ask 'so what'; discuss the different findings rather than restating.

Additional suggestions/edits:

Lines 132-140: All this additional information reads as rationale for the choice of books as the data source. This belongs in the methods.

Line 138/9: Incomplete sentence

Line 156/7: Would like to see a reference to support this claim

Line 183: Use past tense to be consistent

Line 211-13: Would be good to comment on how this book deals with the topic of your research

Inconsistencies throughout regarding the number of homosexual authors in the sample. (Line 232 says there is 2; the table shows only 1 and there are two other occasions where the number is inconsistent)

Line 248: omit 'be'

Line 285: space missing

Line 413: 'an' is missing from the sentence

Line 456: Include the page number for the quote

Line 578 and 579: Both sentences start with 'also'.

Line 594: replace 'but' with 'as'

An additional limitation to consider is that authors were from varying countries. It is likely that the health care systems differ significantly across countries. What are the implications of this?

Line 617/18: This needs updating based on your updated search method.

Line 625: Did reference 40 specifically discuss the reasons for why men write about their eating disorder?

Line 627/8: Revisit this sentence as you don't know whether the findings would corroborate your findings or not.

Author Response

Reviewer 1 The manuscript now offers a lot more context to the reader and depth of analysis which has improved the paper significantly. Major comment: Much of the discussion reads largely as a summary of the results. For example the section on the type of therapy (group v individual) just states the result. There is a need to go the next step to ask 'so what'; discuss the different findings rather than restating. • Author response: Thank you for this suggestion. We have now expanded the Discussion - see lines 561-579. Additional suggestions/edits: Lines 132-140: All this additional information reads as rationale for the choice of books as the data source. This belongs in the methods. • Author response: We have now moved the paragraph to the Methods - see lines 151-159 Line 138/9: Incomplete sentence • Author response: sentence deleted, see line 155-56. Line 156/7: Would like to see a reference to support this claim • Author response: Reframed the sentence as it is our assumption, see lines 147-48 Line 183: Use past tense to be consistent • Author response: Corrected. See line 180 Line 211-13: Would be good to comment on how this book deals with the topic of your research • Author response: Added, see lines 211-213 Inconsistencies throughout regarding the number of homosexual authors in the sample. (Line 232 says there is 2; the table shows only 1 and there are two other occasions where the number is inconsistent) • Author response: This has now been checked and corrected throughout. Line 248: omit 'be' • Author response: Deleted. See line 252 Line 285: space missing • Author response: Corrected. Line 413: 'an' is missing from the sentence • Author response: Added. See line 420 Line 456: Include the page number for the quote • Author response: We checked this. See line 463. Page numbers were also missing from line 375 and 452. These are now added. Line 578 and 579: Both sentences start with 'also'. • Author response: Corrected. See line 594. Line 594: replace 'but' with 'as' • Author response: Corrected – see line 591. An additional limitation to consider is that authors were from varying countries. It is likely that the health care systems differ significantly across countries. What are the implications of this? • Author response: We agree and have added this. See lines 629-632. Line 617/18: This needs updating based on your updated search method. • Author response: Corrected, see lines 616-618. Line 625: Did reference 40 specifically discuss the reasons for why men write about their eating disorder? • Author response: Not directly. Reference 40 discussed issues related to severe psychiatric illness including Anorexia Nervosa. We have rephrased the sentence on line 627-8. Line 627/8: Revisit this sentence as you don't know whether the findings would corroborate your findings or not. • Author response: See lines 637-639. We agree and have removed the sentence.

Reviewer 2 Report

Thank you for your clear responses to the points raised in the initial review. The edits to the manuscript organization and inclusion of the Table are very helpful to the reader. My main concern remains one of methodology as outlined in point #2 of my review. However, I appreciate the authors' response to this concern (also made by the other reviewer), including the inclusion of these concerns in the limitations section of the discussion.

Author Response

Reviewer 2's Comment:

Thank you for your clear responses to the points raised in the initial review. The edits to the manuscript organization and inclusion of the Table are very helpful to the reader. My main concern remains one of methodology as outlined in point #2 of my review. However, I appreciate the authors' response to this concern (also made by the other reviewer), including the inclusion of these concerns in the limitations section of the discussion.

Authors' Response:

We thank the reviewer for their review of the revised manuscript, and are pleased that the changes made are deemed to be acceptable.